# Ictal fMRI: Mapping Seizure Topography with Rhythmic BOLD Oscillations

**DOI:** 10.3390/brainsci12121710

**Published:** 2022-12-13

**Authors:** David Fischer, Otto Rapalino, Matteo Fecchio, Brian L. Edlow

**Affiliations:** 1Division of Neurocritical Care, Department of Neurology, University of Pennsylvania, Philadelphia, PA 19104, USA; 2Department of Radiology, Massachusetts General Hospital and Harvard Medical School, Boston, MA 02129, USA; 3Center for Neurotechnology and Neurorecovery, Department of Neurology, Massachusetts General Hospital and Harvard Medical School, Boston, MA 02114, USA; 4Athinoula A. Martinos Center for Biomedical Imaging, Massachusetts General Hospital and Harvard Medical School, Charlestown, MA 02129, USA

**Keywords:** seizure, fMRI, status epilepticus

## Abstract

Functional magnetic resonance imaging (fMRI) has shown elevations in the blood-oxygen-level-dependent (BOLD) signal associated with, but insensitive for, seizure. Rather than evaluating absolute BOLD signal elevations, assessing rhythmic oscillations in the BOLD signal with fMRI may improve the accuracy of seizure mapping. We report a case of a patient with non-convulsive, right hemispheric seizures who underwent fMRI. Unbiased processing methods revealed a map of rhythmically oscillating BOLD signal over the cortical region affected by seizure, and synchronous BOLD signal in the contralateral cerebellum. High-resolution fMRI may help identify the spatial topography of seizure and provide insights into seizure physiology.

## 1. Introduction

Electroencephalography (EEG) provides high temporal resolution, but limited spatial resolution, which impedes seizure localization [1,2]. Seizure mapping may be improved with functional magnetic resonance imaging (fMRI), which has better spatial resolution than EEG. fMRI assesses brain activity via neurovascular coupling—highly active neurons draw an influx of oxygenated hemoglobin, which fMRI detects as an increase in blood-oxygen-level-dependent (BOLD) signal. The high neuronal activity characteristic of seizures would be expected to trigger an increase in blood flow and, indeed, previous studies have identified regional BOLD signal elevations associated with seizures [3,4,5,6,7,8]. Mapping seizures with fMRI carries many potential advantages beyond increased spatial resolution, as it may detect seizures in deep brain regions and may help evaluate seizures when EEG is not feasible (for example, in cases of scalp incisions, burns or trauma). Improved seizure mapping may help guide targeted therapies, such as resection, deep-brain stimulation, or non-invasive brain stimulation [1,2].

However, to date, fMRI has demonstrated limited sensitivity for seizure mapping [3], perhaps because not all seizing neurons generate a large BOLD signal response. Moreover, attempts to map seizures with fMRI have often required simultaneous EEG [5,9,10,11,12,13], or confined the analysis to a priori regions of interest [3,14]. Such approaches hamper the technique’s clinical utility, given that simultaneous EEG is often not feasible and a priori regions are often uncertain in a clinical setting. There is thus a clinical need to map the spatial topography of seizures in a sensitive and unbiased fashion. Here, rather than evaluating absolute BOLD signal elevations, we used high-resolution fMRI to investigate whether rhythmic, synchronous oscillations in BOLD signal can identify the spatial topography of seizure in a patient with right hemispheric seizures.

## 2. Materials and Methods

### 2.1. Clinical Case Information

As a proof of principle, we identified a patient with non-convulsive status epilepticus—a 51-year-old man with no significant past medical history who, after resuscitation from a cardiac arrest of unclear etiology, remained in a vegetative state. Prompted by left arm twitching, an EEG revealed frequent seizures, manifesting as generalized periodic discharges of 3–5 Hz with highest amplitude in the right hemisphere, under the F4 electrode (Figure 1A). After trials of three anti-epileptic drugs (levetiracetam, fosphenytoin, and clobazam), and three general anesthetics (propofol, midazolam, and ketamine), the left arm twitching subsided, but electrographic seizures persisted, peaking every 1.8–2.3 min (Figure 1B,C). Arterial spin labeling MRI revealed hyperperfusion of the right hemispheric cortex and contralateral cerebellum (Figure 1D), a pattern consistent with seizure [15].

### 2.2. fMRI Methods

While EEG demonstrated ongoing seizures, the patient underwent fMRI as part of a clinical MRI protocol, using a 3T Skyra MRI scanner (Siemens Healthcare, Erlangen, Germany) and a 32-channel head coil. He remained on propofol 20 mcg/kg/min at the time of the scan to ensure safety, comfort and immobility. The fMRI BOLD sequence was obtained over a 10 min duration, with an echo time of 30.3 ms, a repetition time of 1250 ms, a 2 mm^3^ isotropic voxel size and simultaneous multi-slice acquisition (SMS = 4) to optimize spatial and temporal resolution [16]. Using CONN toolbox software [17], images were preprocessed with realignment, structural segmentation, normalization to Montreal Neurological Institute (MNI) space, and smoothing with a 6 mm full-width at half-maximum Gaussian kernel [18]. The artifact rejection tool (ART) was used to reject outlier volumes that met one of the following thresholds: normalized BOLD signal Z ≥ 3.0, absolute subject motion ≥ 0.5 mm, absolute subject rotation ≥ 0.05 radians, scan-to-scan motion ≥ 1.0 mm, or scan-to-scan rotation ≥ 0.02 radians. Images were denoised using the cerebrospinal fluid and white matter principal components as nuisance covariates, with the anatomical component-based noise correction method [19]. Finally, low-frequency BOLD fluctuations were isolated with a low-pass temporal filter (0.008–0.09 Hz).

Two unbiased techniques were used to identify a dominant BOLD rhythm suggestive of seizure: a brain-wide seed-to-voxel analysis using a seed containing all gray matter (segmented with CONN), and an independent components analysis (ICA; 10 factors). The first ICA component was identified as the dominant rhythm, and its peak voxel was identified as the rhythm focus. The BOLD time series was extracted from a 5 mm sphere centered on this focus, which was then used as a seed in a brain-wide seed-to-voxel analysis.

We performed one-sample *t* tests to compare the patient’s imaging metrics to those of seven healthy controls (with no history of neurological, psychiatric, cardiovascular, pulmonary, renal or endocrinological disease) scanned with identical parameters, who provided written consent in an IRB-approved study (https://clinicaltrials.gov/ct2/show/NCT03504709 (accessed on 11 December 2022)).

## 3. Results

The gray matter seed-to-voxel analysis revealed a cluster of synchronous BOLD oscillations in the right hemisphere (Figure 2A). Using averaged Z scores, the ratio of signal in the right hemisphere to signal in the left hemisphere was 1.78. The dominant ICA component showed a similar topography (Figure 2B), and the BOLD signal within its focus (*x* = 56, *y* = −38, *z* = 46; Figure 2C) demonstrated rhythmic spikes every 1.8–2.0 min (a rate similar to that of the electrographic discharges; Figure 2D). Moreover, when this focus was used as a seed in a seed-to-voxel analysis, a synchronous signal was identified in the contralateral cerebellum (Figure 2E; ratio of signal in right versus left cerebellum was 0.85). In contrast, in healthy controls, the gray matter seed-to-voxel analysis revealed a poorly synchronous, diffuse, symmetric signal (Figure 2A; mean ratio of signal in right versus left hemispheres was 1.05 (SD 0.046), significantly less lateralized than that of the patient (*p* < 0.0001)), and the BOLD signal within the focus revealed no discernable rhythmicity (Figure 2D) or lateralized synchrony in the cerebellum (Figure 2E; mean ratio of signal in right versus left cerebellum was 0.94 (SD 0.10), significantly less lateralized than that of the patient (*p* < 0.05)).

## 4. Discussion

Using complementary and unbiased methods for measuring synchronous BOLD oscillations, we identified an fMRI signature of seizure with temporal characteristics similar to that of EEG, but improved spatial resolution. High-resolution fMRI could identify rhythmic BOLD signal oscillations, offering a potentially more sensitive and direct indicator of seizure. This unbiased approach also permitted the localization of seizure without necessitating simultaneous EEG or a priori regions of interest often relied upon in prior fMRI studies, facilitating potential clinical translation.

Moreover, high-resolution fMRI revealed synchronous activity between the cortical region of seizure and the contralateral cerebellum. It is well established that focal seizures affect the contralateral cerebellum, as evidenced by atrophy, edema, or, as in the case of this patient, hyperperfusion of the contralateral cerebellar hemisphere [15,20,21]. These imaging changes have been presumed to be caused by diaschisis—a focal disturbance in one region of the brain caused by a disturbance in a remote but anatomically connected region—given the known corticocerebellar circuits connecting the cortex to the contralateral cerebellum [15,20,21]. However, apart from studies showing ictal hyperperfusion of the contralateral cerebellum, to date the propagation of seizure activity from the cortex to the contralateral cerebellum has not been definitively demonstrated. The synchronous oscillations in BOLD signal observed here, between the cortical region of seizure and the contralateral cerebellum, suggest that such propagation does occur, supporting the theory of diaschisis commonly invoked to explain cerebellar imaging findings in seizure.

This technique may offer opportunities for clinical care. First, ictal fMRI may help identify seizures in deeper structures of the brain—which can be missed by conventional scalp electrodes, contributing to the imperfect sensitivity of scalp EEG [22]—without resorting to invasive depth electrodes. Second, ictal fMRI may help detect seizures when conventional electrode placement is not feasible, as in cases of scalp incisions, burns or trauma. Third, more precise seizure mapping may facilitate the application of targeted therapies, such as resection, deep-brain stimulation, or non-invasive brain stimulation [1,2].

Ictal fMRI is not, however, without limitations. While these findings offer a proof of principle of ictal fMRI, reproducing this technique in larger cohorts is necessary to determine the generalizability of these findings to other patients with different seizure types. The analytic techniques applied here may also be developed and refined. It is possible that dynamic phase coherence or graph theory analysis [23] may further characterize seizure topography and propagation. Regarding the clinical application of ictal fMRI, there are further limitations to consider. Despite potential advantages, ictal fMRI is substantially more costly and complex to obtain and interpret than conventional EEG. Moreover, to ensure patient safety, and given the susceptibility of fMRI to motion artifacts, this approach may be inappropriate for patients with convulsive seizures, or for patients who are otherwise insufficiently stable to tolerate a prolonged scan.

## 5. Conclusions

In summary, these findings illustrate that high-resolution fMRI can be used to detect the BOLD signal manifestations of seizure, potentially providing a more precise seizure topography, and offering insights into seizure physiology. These findings warrant further study and present opportunities for neuroscientific investigation and clinical care.

## Figures and Tables

**Figure 1 brainsci-12-01710-f001:**
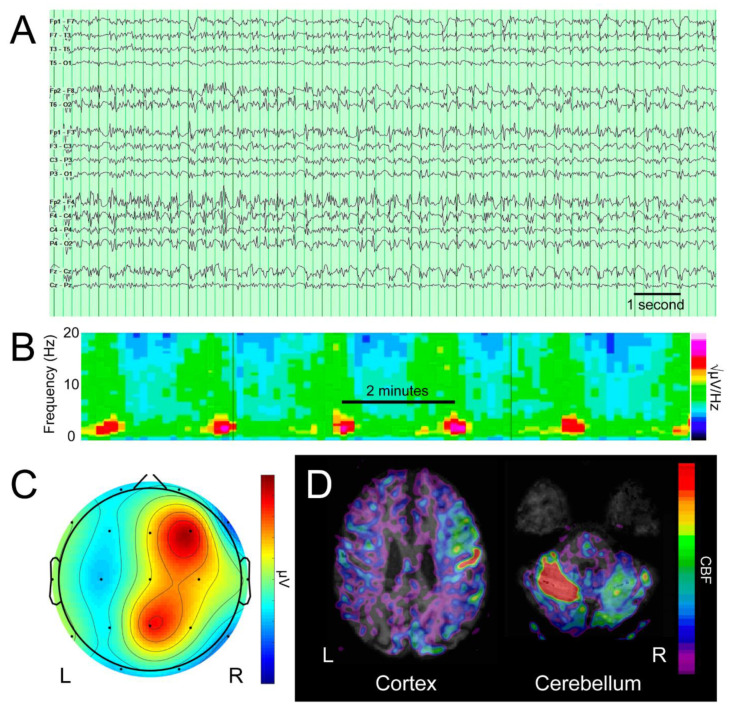
**Clinical data**. The patient’s electroencephalogram (EEG) is depicted in (**A**), the spectrogram of discharge frequency depicted in (**B**), a representative topographic voltage distribution, in a Laplacian montage in (**C**), and arterial spin labeling MRI, revealing increased cerebral blood flow (CBF) in the right cortex and contralateral cerebellum in (**D**).

**Figure 2 brainsci-12-01710-f002:**
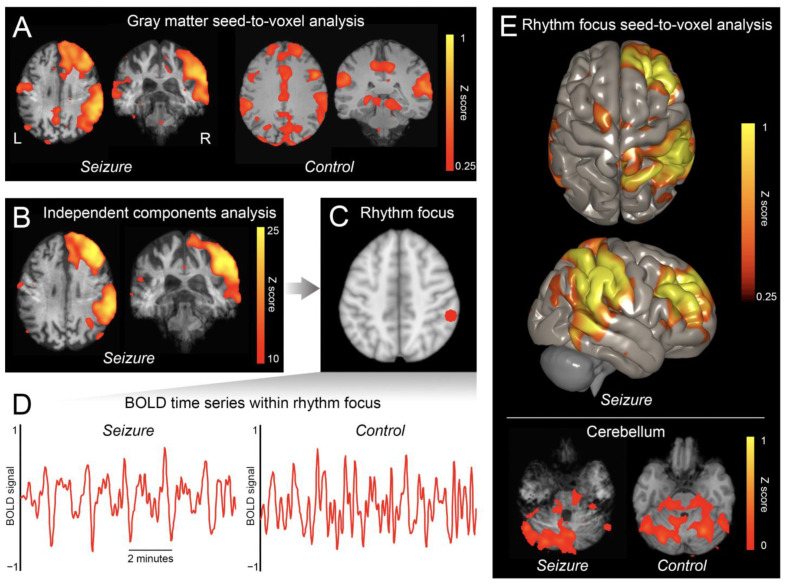
**Ictal fMRI.** Throughout the figure, “Seizure” refers to the patient’s imaging, while “Control” refers to imaging of a representative healthy control. When the gray matter was used as a seed in a seed-to-voxel analysis (**A**), there was a synchronous signal in the patient’s right hemisphere, and a weak, symmetric signal in healthy controls. The first component of an independent components analysis (**B**) revealed a similar topography of signal in the patient’s right hemisphere. The peak voxel of this component was identified as the rhythm focus, and a 5 mm sphere was placed at this coordinate ((**C**); depicted in red). The blood-oxygen-level-dependent (BOLD) time series within this sphere revealed rhythmic oscillations in the patient (**D**), but no discernable rhythmicity in the healthy controls. The rhythm focus was used as a seed in a seed-to-voxel analysis (**E**), demonstrating synchronous signal in the contralateral cerebellum in the patient, but no lateralized synchronous signal in healthy controls (signal masked within cerebellum).

## Data Availability

The data presented in this study are available on request from the corresponding author. The data are not publicly available as they include protected health information.

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
