# Peer review of "Ictal fMRI: Mapping Seizure Topography with Rhythmic BOLD Oscillations"

_brainsci, 2022, doi:10.3390/brainsci12121710_

Round 1
Reviewer 1 Report
1- Introduction could be expanded on the role of Functional magnetic resonance imaging on seizures mapping
2- Medical history for the patient should be included in Material section
3- Criteria for selecting healthy volunteers should be included
Author Response
1- Introduction could be expanded on the role of Functional magnetic resonance imaging on seizures mapping
We have expanded our introduction in three ways. We have added additional clarification on the physiology of how seizures trigger a change in BOLD signal (lines 32-33: “The high neuronal activity characteristic of seizures would be expected to trigger an increase in blood flow” ). We have highlighted the clinical implications of fMRI for seizure mapping (lines 34-39: “Mapping seizures with fMRI carries many potential advantages beyond increased spatial resolution, as it may detect seizures in deep brain regions and may help evaluate seizures when EEG is infeasible (for example, in cases of scalp incisions, burns or trauma). Improved seizure mapping may help guide targeted therapies, such as resection, deep-brain stimulation, or non-invasive brain stimulation”). We have also added additional citations of previous studies evaluating seizure with fMRI (DOI: 10.1016/j.eplepsyres.2016.09.010; DOI: 10.1016/j.nicl.2016.03.010; DOI: 10.1111/epi.12329).
2- Medical history for the patient should be included in Material section
The patient had no past medical history prior to this cardiac arrest of unclear etiology. We have clarified this on lines 63-65: “As a proof of principle, we identified a patient with non-convulsive status epilepticus – a 51-year-old man with no significant past medical history who, after resuscitation from a cardiac arrest of unclear etiology, remained in a vegetative state”.
3- Criteria for selecting healthy volunteers should be included
Healthy controls were defined as subjects with no history of neurological, psychiatric, cardiovascular, pulmonary, renal or endocrinological disease. We have clarified this in lines 109-113: “We performed one-sample t tests to compare the patient’s imaging metrics to those of seven healthy controls (with no history of neurological, psychiatric, cardiovascular, pulmonary, renal or endocrinological disease) scanned with identical parameters, who provided written consent in an IRB-approved study (https://clinicaltrials.gov/ct2/show/NCT03504709)”.
Reviewer 2 Report
An interesting and innovative article. As the authors note, there are limitations to the study in patients with tonic-clonic seizures. Figure D is unclear. Please explain what control means - the right side of the figure. There is no proper bioelectrical activity of the brain in the picture - i.e. alpha waves. Please correct this and explain this part in more detail.
Author Response
An interesting and innovative article. As the authors note, there are limitations to the study in patients with tonic-clonic seizures. Figure D is unclear. Please explain what control means - the right side of the figure. There is no proper bioelectrical activity of the brain in the picture - i.e. alpha waves. Please correct this and explain this part in more detail.
We thank Reviewer 2 for their kind words. We have made further clarifications to Figure 2 in response to these comments. We have added a title to panel D of Figure 2 to clarify its contents – we display the BOLD signal time series within the rhythm focus both for the patient and for a representative control, demonstrating rhythmicity in the former but no rhythmicity in the latter. We have clarified in the Figure 2 legend that “Control” refers to data collected from a representative healthy control. We have also removed a sentence from the discussion that may have contributed to the confusion about Figure 2D’s contents. We hope that these clarifications address the Reviewer’s question about alpha waves – because the data represent rs-fMRI data (derived from BOLD signal), whereas alpha waves are comprised of electrical activity on EEG, the data depicted in Figure 2 would not be expected to show alpha waves.
Reviewer 3 Report
Major issues
#1. At the end of Introduction, I suggest to say, “for a patient with right frontal lobe epilepsy”.
#2. Since the patient exhibited clinical manifestation as left arm twitching, NCSE is inappropriate. “Epilepsia Partialis Continua” might be better to use.
#3. Line 49, you may say “under the right frontal electrodes” without specifying the F4.
#4. Some similar case reports are following.
DOI: 10.1016/j.eplepsyres.2016.09.010
DOI: 10.1016/j.nicl.2016.03.010
DOI: 10.1111/epi.12329
Author Response
Major issues
#1. At the end of Introduction, I suggest to say, “for a patient with right frontal lobe epilepsy”.
We have included additional qualification to the final sentence of the introduction on line 60 as follows: “Here, rather than evaluating absolute BOLD signal elevations, we used high-resolution fMRI to investigate whether rhythmic, synchronous oscillations in BOLD signal can identify the spatial topography of seizure in a patient with right hemispheric seizures.” We changed “right frontal lobe” to “right hemispheric” as it is not clear from the combination of EEG and fMRI data whether the focus is in the frontal lobe or parietal lobe. We changed “epilepsy” to “seizures” to draw attention to the fact that the patient was actively having seizures, and did not have merely have a propensity for seizures.
#2. Since the patient exhibited clinical manifestation as left arm twitching, NCSE is inappropriate. “Epilepsia Partialis Continua” might be better to use.
Although the patient initially had left arm twitching, by the time of the fMRI scan the twitching had subsided, prompting the use of the term NCSE. We have added further clarification on this point to lines 68-71: “After trials of three anti-epileptic drugs (levetiracetam, fosphenytoin, and clobazam) and three general anesthetics (propofol, midazolam, and ketamine), the left arm twitching subsided, but electrographic seizures persisted, peaking every 1.8 – 2.3 minutes (Fig 1B, Fig 1C)”.
#3. Line 49, you may say “under the right frontal electrodes” without specifying the F4.
The fMRI, which has superior spatial resolution to EEG, suggests that the actual seizure focus may have been more posterior, in the parietal lobe. To avoid confusion, we chose to describe which electrode the seizure focus was closest to, allowing fMRI to provide additional precision with regard to the exact location of the seizure focus.
#4. Some similar case reports are following.
DOI: 10.1016/j.eplepsyres.2016.09.010
DOI: 10.1016/j.nicl.2016.03.010
DOI: 10.1111/epi.12329
We appreciate Reviewer 3’s referrals to other similar case reports, and for the opportunity to describe how our case differs from previous studies. We note that the first case report (10.1016/j.eplepsyres.2016.09.010) involved use of a “seed region”, or an a priori region of interest, as the foundation of their analysis, and the second two case reports (DOI: 10.1016/j.nicl.2016.03.010 and DOI: 10.1111/epi.12329) used simultaneous EEG during the fMRI. As we note in our introduction, these additions substantially limit the clinical feasibility of these techniques, as a priori regions are often unknown, and simultaneous EEG is often infeasible. We have highlighted in our introduction that in this study, we used unbiased analysis methods that do not rely on simultaneous EEG, which we hope will facilitate the clinical implementation of this technique. We have added these three references to our introduction, as examples of prior studies that have relied on a priori regions of interest and simultaneous EEG.
Reviewer 4 Report
Abstract: first sentence might be revised and improved, as regard orthography and contents. You may explain better the relationship between seizures and BOLD signal
Introduction: You may consider to increase your description of fMRI advantages in localizing epileptic region. Last sentence might be improved, you could underline better the aim of the study.
Methods: sentence 48 – You should describe better the EEG trace, you might use American Clinical Neurophysiology Society’s Standardized Critical Care EEG Terminology: 2021 Version
Line 48 – as for line 46
You might separate the clinical case from the description of methods.
Conclusion: you should underline better the relevance of your study and all the possible clinical application of fMRI in patients with epilepsy
Author Response
Abstract: first sentence might be revised and improved, as regard orthography and contents.
We have reworded the first sentence to try to improve its clarity, on lines 27-28, as follows: “Electroencephalography (EEG) provides high temporal resolution, but limited spatial resolution, which impedes seizure localization”.
You may explain better the relationship between seizures and BOLD signal
We have added further clarification of this relationship on lines 32-33, as follows: “The high neuronal activity characteristic of seizures would be expected to trigger an increase in blood flow, and indeed, previous studies have identified regional BOLD signal elevations associated with seizures.”
Introduction: You may consider to increase your description of fMRI advantages in localizing epileptic region.
We have added further description of the advantages of ictal fMRI in the introduction, on lines 34-39, as follows: “Mapping seizures with fMRI carries many potential advantages beyond increased spatial resolution, as it may detect seizures in deep brain regions and may help evaluate seizures when EEG is infeasible (for example, in cases of scalp incisions, burns or trauma). Improved seizure mapping may help guide targeted therapies, such as resection, deep-brain stimulation, or non-invasive brain stimulation”
Last sentence might be improved, you could underline better the aim of the study.
We have added further clarity on the specific aim of this study by adjusting the lines 46-60 as follows: “Here, rather than evaluating absolute BOLD signal elevations, we used high-resolution fMRI to investigate whether rhythmic, synchronous oscillations in BOLD signal can identify the spatial topography of seizure in a patient with right hemispheric seizures.”
Methods: sentence 48 – You should describe better the EEG trace, you might use American Clinical Neurophysiology Society’s Standardized Critical Care EEG Terminology: 2021 Version
Line 48 – as for line 46
We have added further detail to our description of the EEG, informed by the terminology guidelines suggested by the Reviewer, on lines 65-71 as follows: “Prompted by left arm twitching, an EEG revealed frequent seizures, manifesting as generalized periodic discharges of 3-5 Hz with highest amplitude in the right hemisphere, under the F4 electrode (Fig 1A). After trials of three anti-epileptic drugs (levetiracetam, fosphenytoin, and clobazam) and three general anesthetics (propofol, midazolam, and ketamine), the left arm twitching subsided, but electrographic seizures persisted, peaking every 1.8 – 2.3 minutes (Fig 1B, Fig 1C).”
You might separate the clinical case from the description of methods.
Subheadings were added to the Materials and Methods section to separate the clinical case from the description of the fMRI methods, as suggested.
Conclusion: you should underline better the relevance of your study and all the possible clinical application of fMRI in patients with epilepsy
We have described the clinical applications that we could think of, and that we felt were justified by this study’s findings. To help highlight these potential applications, we have reiterated them in the introduction. If there are further applications justified by these findings that we have not included, we are open to including others.
Round 2
Reviewer 3 Report
I endorse this version.